# Optimal Design of Galvanic Vestibular Stimulation for Patients with Vestibulopathy and Cerebellar Disorders

**DOI:** 10.3390/brainsci13091333

**Published:** 2023-09-16

**Authors:** Thanh Tin Nguyen, Seung-Beop Lee, Jin-Ju Kang, Sun-Young Oh

**Affiliations:** 1Department of Neurology, Jeonbuk National University Hospital, Jeonbuk National University School of Medicine, Jeonju 54907, Republic of Korea; thanhtin167@gmail.com (T.T.N.); jinj_k@naver.com (J.-J.K.); 2Department of Pharmacology, Hue University of Medicine and Pharmacy, Hue University, Hue 49120, Vietnam; 3School of International Engineering and Science, Graduate School of Integrated Energy-AI, Jeonbuk National University, Jeonju 54896, Republic of Korea; seungbeop.lee@jbnu.ac.kr; 4Research Institute of Clinical Medicine of Jeonbuk National University-Biomedical Research Institute of Jeonbuk National University Hospital, Jeonju 54907, Republic of Korea

**Keywords:** galvanic vestibular stimulation, vertigo, imbalance, optimal design, sensitivity analysis, vestibulopathy, cerebellar ataxia

## Abstract

Objectives: Galvanic vestibular stimulation (GVS) has shown positive outcomes in various neurological and psychiatric disorders, such as enhancing postural balance and cognitive functions. In order to expedite the practical application of GVS in clinical settings, our objective was to determine the best GVS parameters for patients with vestibulopathy and cerebellar disorders using optimal design calculation. Methods: A total of 31 patients (26 males, mean age 57.03 ± 14.75 years, age range 22–82 years) with either unilateral or bilateral vestibulopathy (n = 18) or cerebellar ataxia (n = 13) were enrolled in the study. The GVS intervention included three parameters, waveform (sinusoidal, direct current [DC], and noisy), amplitude (0.4, 0.8, and 1.2 mA), and duration of stimulation (5 and 30 min), resulting in a total of 18 GVS intervention modes as input variables. To evaluate the effectiveness of GVS, clinical vertigo and gait assessments were conducted using the Dizziness Visual Analogue Scale (D-VAS), Activities-specific Balance Confidence Scale (ABC), and Scale for Assessment and Rating of Ataxia (SARA) as output variables. Optimal design and local sensitivity analysis were employed to determine the most optimal GVS modes. Results: Patients with unilateral vestibulopathy experienced the most favorable results with either noisy or sinusoidal GVS at 0.4 mA amplitude for 30 min, followed by DC GVS at 0.8 mA amplitude for 5 min. Noisy GVS at 0.8 or 0.4 mA amplitude for 30 min demonstrated the most beneficial effects in patients with bilateral vestibulopathy. For patients with cerebellar ataxia, the optimal choices were noisy GVS with 0.8 or 0.4 mA amplitude for 5 or 30 min. Conclusions: This study is the first to utilize design optimization methods to identify the GVS stimulation parameters that are tailored to individual-specific characteristics of dizziness and imbalance. A sensitivity analysis was carried out along with the optimal design to offset the constraints of a limited sample size, resulting in the identification of the most efficient GVS modes for patients suffering from vestibular and cerebellar disorders.

## 1. Introduction

Galvanic vestibular stimulation (GVS) is a non-invasive method used to stimulate the vestibular system and has long been considered a useful tool for investigating the vestibular and balance systems [1,2]. GVS activates both primary otolithic and semicircular canal neurons and their cortical projections to vestibular cortex [2,3,4]. The firing rates of peripheral vestibular afferents are increased by galvanic currents at the cathode and decreased at the anode [4,5,6], implying excitatory stimulation at the cathode and inhibitory stimulation at the anode, and generating a series of action potentials of the vestibular sensory organ [2,7,8]. Accordingly, GVS has been shown to alter the vestibular commissural inhibitory system to improve a variety of vestibular-related functional deficits, including not only motor coordination and posture but also cognitive and memory impairments [9]. Given the substantial afferent and efferent connections that exist between the vestibular nuclei and the cerebellum, particularly the posterior cerebellum [10], along with emerging evidence of the positive effects of non-invasive cerebellar stimulation in treating cerebellar disorders [11], there is a theoretical basis for suggesting that GVS could potentially enhance cerebellar function and address cerebellar dysfunctions. Recent studies have demonstrated the beneficial effects of GVS on improving body balance [12,13,14], gait [15], dynamic walking [16], spatial learning [17], executive memory [18], and visual memory [19]. However, inconsistency in GVS intervention paradigms as well as a lack of large-scale clinical trials have kept GVS from being widely used in real clinical practice. The effects of GVS can vary significantly depending on its parameters, including waveforms (direct current [DC], sinusoid, or noisy), amplitudes, frequencies, application times, and the number of sessions [7]. For example, subthreshold GVS had a positive effect on postural stability and spatial memory performance [12,17,20], whereas high amplitude of GVS had a negative impact on postural and cognitive performances [21]. In our previous study, we investigated the vestibular, cutaneous perceptual, and oculomotor thresholds to DC-GVS current and proposed a method for determining the GVS threshold of subjects based on testing vestibular perception [22].

In our study, we aimed to optimize GVS parameters for both vestibulopathy and cerebellar disorder patients, despite their differing pathophysiology but due to their similar symptoms of imbalance and dizziness. While numerous published reports have demonstrated the beneficial effects of GVS on balance performance, fundamental questions remain regarding the optimal GVS parameters. To accelerate the clinical application of GVS, our study aimed to identify the most effective GVS mode for patients with vestibulopathy and cerebellar ataxia by assessing three core factors: waveform, amplitude, and duration. In these patients, whose primary symptoms manifest as imbalance and dizziness, we employed clinical metrics including the Dizziness Visual Analogue Scale (D-VAS), the Activities-specific Balance Confidence Scale (ABC), and the Scale for Assessment and Rating of Ataxia (SARA). We used an algorithm-based optimal design for model predictions to understand the correlation between GVS parameters and their subsequent effects, especially in resource-constrained scenarios. Optimal design, which maximizes information yield while taking into account budget constraints that limit the resources available for the study, has recently emerged as a valuable statistical methodology for designing a wide range of studies [23]. It provides a systematic, quantitative approach to selecting study units (i.e., patient groups) in the most informative manner for observational studies and assigning study units to intervention conditions in the most informative manner for experimental studies [23]. This novel approach has been widely used in clinical research, particularly in studies exploring a typical pattern of pharmacokinetics over time, which has assisted in reducing the number of sampling times, improving existing therapies or diagnostics, and providing recommendations for effective dose regimen [24,25,26,27,28,29]. The optimal design approach provides a useful framework for evaluating the sensitivity of design decisions to deviations from usual assumptions [23]. In addition, using a sensitivity analysis (SA) approach, the identification of key design parameters is implemented in design optimization [30]. SA is a system identification process that is carried out to construct a mathematical model based on input–output data measured in the real system, with steps including model structure selection, design of experiments, data collection, parameter optimization, and model validation [31,32].

The goal of this study was to identify the optimal parameters of GVS for alleviating symptoms of dizziness and imbalance in patients with vestibulopathy and cerebellar ataxia based on an optimal design calculation, which has not been used in the clinical application of GVS. We selected three cardinal factors of GVS settings (waveform, amplitude, and duration) as input variables for the modeling process and clinical dizziness and gait scales as output variables.

## 2. Methods

### 2.1. Study Design and Participants

This study included a cohort of patients diagnosed with uni- or bilateral vestibulopathy (UVP and BVP) and cerebellar ataxia who visited Jeonbuk National University Hospital between September 2021 and August 2022 (n = 31; mean age 57.03 ± 14.75 years; age range 22–82 years; 26 males) (Table 1). Vestibular disorders were confirmed through clinical examinations performed by a senior neurotologist (S.Y. Oh) and vestibular function tests including a video head impulse test (vHIT), bithermal caloric test and ocular and cervical vestibular evoked myogenic potentials (cVEMP and oVEMP). Vestibular function tests were described in previous reports [33,34]. Nine patients with acute or chronic UVP and nine BVP patients, including patients with idiopathic bilateral vestibulopathy (n = 2), presbyvestibulopathy (n = 3) and CANVAS (cerebellar ataxia, neuropathy and vestibular areflexia syndrome, n = 4), were recruited. We also included 13 patients with cerebellar ataxia, including multiple system atrophy, cerebellar subtype (MSA-C) (n = 3), spinocerebellar ataxia (SCA, n = 3) and late-onset cerebellar ataxia (n = 7) (Table 1). To ensure the accuracy of self-assessment responses, general cognitive function was evaluated using the Mini–Mental State Examination (MMSE) and only patients with a score higher than 27 were included. The output was formed based on clinical assessments reflecting dizziness perception and imbalance such as the Dizziness Visual Analogue Scale (D-VAS) [35], Activities-specific Balance Confidence Scale (ABC), and Scale for Assessment and Rating of Ataxia (SARA) (Figure 1). Each assessment session commenced five minutes after the cessation of the stimulation and typically lasted between 10 to 15 min. These data were then processed in accordance with the experimental design depicted in Figure 2.

All participants provided informed consent and received monetary compensation for their participation. Experiments were reviewed and approved by the Institutional Review Board at Jeonbuk National University Hospital (no. 2021-07-013).

### 2.2. Input Variations

Data were collected regarding the waveforms, amplitudes, and durations of GVS interventions and used to create a sample of input variables for the formula. As described in our recent study [22], GVS was delivered using a CE-certified battery-driven constant current stimulator (neuroConn DC-Stimulator Plus; neuroConn, Ilmenau, Germany) via a pair of 35 cm^2^ rectangular conductive rubber electrodes (5 × 7 cm; the maximum current density in this study was estimated to be 57.14 μA/cm^2^, corresponding to a charge density of 1.71 Coloumb/cm^2^ at the skin surface) (neuroConn) coated with electrode gel and placed binaurally over both mastoids. Participants were seated in a comfortable chair equipped with armrests, located in a soundproof and dimly lit room. The GVS comprised three different levels: sinusoidal (1 Hz), noisy (low-frequency noise LF, 0–100 Hz), and DC signals. In the UVD group, all three protocols (sinusoidal, noisy, and DC with cathode in lesion side) were applied, whereas only sinusoidal and noisy modes were used in the BVD and cerebellar groups. The amplitude of the intervention varied between 0.4, 0.8, and 1.2 mA, with intervention durations of 5 or 30 min, resulting in a total of 18 GVS intervention parameters labeled as modes 1 to 18 (Figure 1). In this study, from the multitude of possible combinations of waveform, amplitude, and duration, we opted for 18 modes as a pragmatic approach for an initial exploration in a clinical context. We used three waveforms: DC, sinusoidal, and noisy. Regarding amplitude, we took into account the vestibular perceptual threshold of GVS. Although there is no standardized threshold, many clinical studies accept a GVS threshold of 1mA [19,22,36,37,38,39]. Therefore, we chose amplitudes of 0.4 mA (subthreshold), 0.8 mA (around threshold), and 1.2 mA (suprathreshold). We also utilized short and long GVS durations of 5 min and 30 min, respectively. However, patients with UVD were exposed to all 18 modes, while those with BVD or cerebellar ataxia were given only 12 modes, excluding the DC current. The DC mode may not be apt for bilateral conditions like BVD or cerebellar ataxia due to its polarization effects. The mode order was randomized using Microsoft Excel’s = Rand() function. To minimize the learning effect on GVS trial results and ensure the observed clinical changes were due to GVS interventions, we conducted fewer than three GVS sessions a day with at least 30 min intervals between them.

### 2.3. Run GVS Application Models

#### 2.3.1. Original Raw Output

To assess output data, we evaluated three scales reflecting vestibular and balance functions: D-VAS, ABC, and SARA (Figure 1). D-VAS was utilized to measure the severity of dizziness on a scale of 0 to 10, where 0 indicated no symptoms and 10 indicated the most severe symptoms imaginable [35]. The Korean version of the ABC ranging from 0 to 100% was calculated as the mean of all ratings of 16 activities, with each activity rated by patients on a visual scale ranging from 0% to 100% as to how confident they felt they would not lose their balance or become unsteady [40]. SARA, an 8-item clinical rating scale ranging from 0 (no ataxia) to 40 (severe ataxia), was used to determine the severity of ataxia (most severe ataxia) [41]. D-VAS and SARA values both positively correlate with the severity of dizziness and postural instability, whereas the ABC value negatively correlates with severity. Each assessment was determined before the stimulation (at baseline) and commenced five minutes after the cessation of the stimulation and typically lasted between 10 and 15 min.

#### 2.3.2. Output Data Normalization

The process of casting the data to a specific range between 0 and 1, known as Min–Max normalization, was used to suit the consistency and eliminate the large differences in the ranges of the raw values of D-VAS, ABC, and SARA. Normalization was accomplished by determining the maximum (Xmax) and minimum (Xmin) values of each variable across all data sets and then estimating each normalized variable of D-VAS, ABC, and SARA from the original raw values (Figure 2) using the following formula [42,43,44]: Xnorm = (X − Xmin)/(Xmax − Xmin).

#### 2.3.3. Output Data Transformation

The estimated normalized data of D-VAS, ABC, and SARA prior to and after GVS intervention were transformed and consolidated into forms suitable for performing specific data mining tasks using the following formulas:Norm(Δ D-VAS) = Norm(D-VAS _post_) − Norm(D-VAS _pre_)
Norm(Δ ABC) = Norm(ABC _post_) − Norm(ABC _pre_)
Norm(Δ SARA) = Norm(SARA _post_) − Norm(SARA _pre_),
where Norm(Δ D-VAS), Norm(Δ ABC), and Norm(Δ SARA) are the indices representing the changes in D-VAS, ABC, and SARA values, respectively; Norm(D-VAS _pre_), Norm(ABC _pre_), and Norm(SARA _pre_) are estimated normalized data of D-VAS, ABC, and SARA values prior to GVS application; and Norm(D-VAS _post_), Norm(ABC _post_), Norm(SARA _post_) are estimated normalized data of D-VAS, ABC, and SARA values after GVS application.

#### 2.3.4. Output Data Integration

During this process, the efficacy of GVS, which had previously been assessed separately using D-VAS, ABC, and SARA, was combined into a single parameter: F value. This value was estimated using the following formula:F = Norm(Δ D-VAS) − Norm(Δ ABC) + Norm(Δ SARA)
where Norm(Δ D-VAS), Norm(Δ ABC), and Norm(Δ SARA) are the indices representing the changes in D-VAS, ABC, and SARA values, respectively.

#### 2.3.5. Building the Ranking Orders

This step calculated the sum of F values (F_sum_) for each GVS mode for each disease-categorized subgroup, which includes uni- or bilateral vestibulopathy and cerebellar ataxia. For each subgroup, an ascending F_sum_ value ranking reflecting the priorities of GVS modes was constructed to allow for the selection of modes with a greater positive effect on improving vestibular perception and gait performance (Figure 2).

### 2.4. Sensitivity Analysis

A sensitivity analysis (SA) approach was implemented in design optimization for the identification of key design parameters. The SA approach constructs a mathematical model based on input–output data measured in the real system, with steps including model structure selection, design of experiments, data collection, parameter optimization, and model validation [30,31,32]. SA was theoretically required as a prerequisite and an accuracy criterion in model design for both diagnostic and prognostic studies [45]. Global SA, which considers the effect of input uncertainty across the entire input space, is thought to be a more reliable but computationally demanding method [46]. In the current study, we utilized local SA, also known as the one-factor-at-a-time method, which focuses on the effects of uncertain inputs around a point (or base case), due to its simple implementation and easy interpretation [46].

Output data integration (F value) was used to quantify the degree of sensitivity with respect to input parameter variations. To categorize input parameters based on their sensitivity, the sensitivity coefficient (SC) was calculated using the following formula:SC_D-VAS_ = F_sum_/Norm(Δ D-VAS)
SC_ABC =_ F_sum_/Norm(Δ ABC)
SC_SARA =_ F_sum_/Norm(Δ SARA)
where F_sum_ is the sum of F values for each GVS mode and Norm(Δ D-VAS), Norm(Δ ABC), and Norm(Δ SARA) are the indices representing the changes in D-VAS, ABC, and SARA values, respectively. For each output, its contribution to building the model becomes more important as its partial absolute value of SC approaches 1.

### 2.5. Statistical Analysis: 

Data were compiled and analyzed using SPSS Statistics version 23.0 (IBM Corp., Armonk, NY, USA). The non-parametric variables were indicated as a median (95% confidence interval [CI]). All the tests were performed at a 0.05 level of significance.

## 3. Results

### 3.1. Clinical Characteristics

The demographic and clinical characteristics of 31 patients with vestibular and cerebellar disorders are listed in Table 1. There were 18 patients with vestibular disorders (UVP, n = 9 and BVP, n = 9) and 13 patients with cerebellar ataxia (multiple system atrophy, cerebellar subtype (MSA-C), n = 3; spinocerebellar ataxia (SCA), n = 3; and idiopathic late-onset cerebellar ataxia, n = 7). Patients with unilateral vestibulopathy showed ipsilateral caloric weakness (median = 39.44%, 95%CI = 29.54–65.3), pathologic vHIT with decreased gain and corrective saccades on the lesion side (ipsilesional gain median = 0.52, 95%CI = 0.27–0.85; contralesional gain median = 0.97, 95%CI = 0.62–1.14) or ipsilateral VEMPs abnormalities (oVEMP asymmetry ratio = 23%, 95%CI = 12–43; cVEMP asymmetry ratio = 21%, 95%CI = 6–37) (Table 1). Patients with bilateral vestibulopathy exhibited decreased caloric weakness bilaterally and decreased gain and corrective saccades on bilateral vHIT. Patients with cerebellar ataxia showed normal caloric response (median = 15.5, 95%CI = 8–19.77), vHIT (right gain median = 1.01, 95%CI = 0.91–1.07; left gain median = 0.99, 95%CI = 0.91–1.04), and VEMPs (oVEMP asymmetry ratio = 11%, 95%CI = 4–36; cVEMP asymmetry ratio = 8%, 95%CI = 4–23). All patients showed normal MMSE scores (median = 29, 95%CI = 28–30, range = 27–30) (Table 1).

Participants were instructed to report any discomfort experienced during the GVS intervention sessions. A handful of patients described sensations of spinning or tilting, as well as mild tingling at the mastoid processes. These mild sensations, primarily observed at the higher DC stimulation level of 1.2 mA, subsided after the stimulation was discontinued (Table 2, Table 3 and Appendix A) [2,7,22].

### 3.2. Determining the Optimal GVS Mode Specific for Vestibulopathy

Table 2 presents a ranking of the GVS parameters that showed positive effects on D-VAS, ABC and SARA in nine patients with UVP. From our analysis, patients with UVP exhibited the best outcomes when subjected to a mode of noisy GVS with 0.4 mA amplitude for 30 min, which has the lowest F_sum_ value (F_sum_ = −0.8; minimum F = −0.65; SC_D-VAS_ = 2.13; SC_ABC_ = −9.51; SC_SARA_ = 2.35). The mode of sinusoidal GVS with 0.4 mA amplitude for 30 min, which has a F_sum_ value of −0.69 (minimum F = −0.39; SC_D-VAS_ = 1.58; SC_ABC_ = −7.38; SC_SARA_ = 4.32) was the second best, followed by GVS with DC stimulation with the cathode on the lesion side, with 0.8 mA amplitude for 5 min (F_sum_ value, −0.64; minimum F = −0.41; SC_D-VAS_ = 2.56; SC_ABC_ = −4.26; SC_SARA_ = 2.67). The sensitivity analysis revealed that these GVS modes had positive effects on all three output parameters (Table 2).

In the case of BVP patients (n = 9), the mode of noisy GVS with 0.8 mA amplitude for 30 min with a F_sum_ value of −0.38 (minimum F = −0.38; SC_D-VAS_ = 1.75; SC_ABC_ = −2.33; SC_SARA_ = undefined) was identified as the most effective (Table 2). Noisy GVS with 0.4 mA amplitude for 30 min, which had the lowest F_sum_ value (F_sum_ = −0.22; minimum F = −0.14; SC_D-VAS_ = undefined; SC_ABC_ = −1; SC_SARA_ = undefined) followed. Sinusoidal GVS mode with 0.4 mA amplitude for 5 min with a F_sum_ value of −0.01 (minimum F = −0.01; SC_D-VAS_ = 1; SC_ABC_ = undefined; SC_SARA_ = undefined) was the third option. The sensitivity analysis revealed they had beneficial effects on D-VAS and ABC scores.

### 3.3. Determining the Optimal GVS Mode Specific for Cerebellar Ataxia

Table 3 and Appendix A shows a ranking of GVS applications for improving dizziness (D-VAS) and imbalance (ABC and SARA) scales in patients with cerebellar disorders (n = 13). The mode of noisy GVS with 0.8 mA amplitude for 5 min, which has the lowest F_sum_ value (F_sum_ = −0.98; minimum F = −0.94; SC_D-VAS_ = 1.3; SC_ABC_ = −6.66; SC_SARA_ = 12.21), was the most effective parameter. The mode of noisy GVS with 0.4 mA amplitude for 30 min with a F_sum_ value of −0.88 (minimum F = −0.38; SC_D-VAS_ = 1; SC_ABC_ = 16.6; SC_SARA_ = 15.69) was the second best for cerebellar ataxia, followed by noisy GVS with 1.2 mA amplitude for 5 min with a F_sum_ value of −0.63 (minimum F = −0.54; SC_D-VAS_ = 1.01; SC_ABC_ = −126; SC_SARA_ = undefined). The sensitivity analysis of these modes revealed beneficial effects on the output variables.

## 4. Discussion

Here, we successfully implemented the optimal design method for the most effective clinical application of GVS using three cardinal factors, waveform, amplitude, and duration, as input variables for the modeling process and clinical scales as output variables. We proposed several options of GVS parameters that showed immediate effects on dizziness perception (D-VAS) and imbalance (ABC and SARA) in patients with vestibular and cerebellar disorders by way of computation of optimal design. From our analysis, patients with UVP exhibited the best outcomes when subjected to either noisy or sinusoidal GVS at an amplitude of 0.4 mA for 30 min. Another promising approach for UVP patients involved the use of DC GVS, with the cathode aligned to the lesion side and an amplitude of 0.8 mA for a brief period of 5 min (Table 2). In the case of BVP patients, noisy GVS, particularly at amplitudes of 0.8 or 0.4 mA over 30 min, was identified as the most effective. For patients diagnosed with cerebellar ataxia, our findings prominently featured noisy GVS. When applied at amplitudes of 0.8 or 0.4 mA for either 5 or 30 min, it produced the most favorable outcomes (Table 3). The sensitivity analysis for each parameter revealed positive effects on the D-VAS, ABC, and SARA.

The effects of GVS are stimulus-dependent; however, how to optimize stimulation parameters to maximize beneficial effects is unknown. In the case of UVP patients, we discovered that all three GVS waveforms (noisy, sinusoidal, and DC) contributed to the best options. However, with GVS thresholds of 1 mA for the vestibular, cutaneous, and oculomotor thresholds [22], subthreshold currents of 0.4 or 0.8 mA were sufficient to elicit a maximal output effect on dizziness and imbalance symptoms. This means that these subthreshold current amplitudes could be extensively used to improve clinical performance while avoiding patient discomfort. A lower current (0.4 mA) with a longer duration application (30 min) with noisy or sinusoidal modes appeared to be more effective than a slightly higher current (0.8 mA) with a shorter application time (5 min) with DC mode in UVP patients (Table 2). In patients with BVP and cerebellar ataxia, noisy GVS at subthreshold amplitudes (0.4 or 0.8 mA) also demonstrated its superiority for the improvement of vertigo and imbalance symptoms compared with sinusoidal waveform (Table 2 and Table 3). The outperforming effects of noisy GVS were thought to be related to stochastic resonance (SR), a phenomenon in non-linear systems in which an appropriate level of noise signal can upgrade a weak subthreshold sensory input to exceed a predefined threshold for causing effects on the signal transmission [47,48]. The beneficial effect of subthreshold, imperceptible noisy GVS on patients with various neurological symptoms has been extensively documented in previous clinical studies [12,15,17,48,49,50,51]. However, the “optimal design” methodology encompasses a plethora of parameters to assess and prioritize GVS efficacy, and thus, pinpointing congruent results for a direct comparison has proven challenging. To pinpoint the optimal combinations, referred to as the “best triads”, derived from the matrix of waveform, intensity (amplitude), and duration, the biophysiological effects of each triad should be thoroughly investigated using functional neuroimaging techniques. To date, the interplay between the effectiveness of GVS and each waveform, intensity, and duration remains uncharted territory. Our hypothesis posits that each triad uniquely amplifies a synergistic influence on the modulation of vestibular afferent firing rates. In terms of intensity, stronger DC currents will cause a more pronounced initial shift in neural activity. However, continuous stimulation (DC) can lead to a steady-state activation or adaptation of the vestibular nerve fibers. After an initial change in activity, the neurons may adapt, reducing their firing rate over time even if the stimulus continues, while the intensity and frequency of the sinusoid will directly influence which neuronal populations are activated. Notably, irregular neurons tend to have a lower activation threshold than regular neurons, as well as different resonance frequencies [52,53]. The duration of the stimulus defines how long this neuronal modulation lasts, and it is believed that adaptation may be slower compared to DC stimulation. Given what we currently know, we speculate that each triad has its own unique combined effect on the modulation of both irregular and regular vestibular afferent firing rates. Such a synergistic impact is deemed beneficial if it is potent enough to cause meaningful charge modulation in the vestibular cortices, thus inducing a neurobiological effect. At the same time, it is crucial to avoid excessive charge buildup in the tissues, as this could be detrimental and potentially cause neuronal damage [7].

Optimal design is a novel study methodology that has been widely used in clinical research, especially for understanding typical patterns of pharmacokinetics over time. It has demonstrated benefits in reducing the number of sampling times (lowering study costs), improving existing therapies or diagnostics, providing recommendations for efficient dose regimens, and extrapolating results from adult trials to pediatric populations (bridging population studies) [24,25,26,27,28,29]. Therefore, for the given resource constraints, the optimization of a clinical trial design that extracts maximum information from the trial while minimizing the sample size of study subjects or their exposure to suboptimal treatment regimens or interventions can assist investigators in efficiently achieving the study objectives [24]. Our goal in the current study was to find the design that maximizes the effectiveness of GVS for the given modes and sample size based on a hypothetical single-step multiple comparisons procedure [26]. To the best of our knowledge, no optimization methods for applicable designs of GVS interventions have yet been implemented. We proposed a simplified optimal design trial, which was presented as a procedure based on a customized F_sum_ value ranking. These rankings enable a flexible and continuous update process based on add-on data when the sample size is constantly increasing throughout the clinical trial in a closed-loop adjusted manner. The flexibility of these rankings allows for the selection of optimal choices among many options, but it also means that each given rating is dynamic and uncertain before a sufficiently large sample is reached. This model-based experimental design is used to exploit the information embedded in the experimental data as soon as it is available and adjust the clinical tests accordingly while it is running [27]. Therefore, the proposed approach based on advanced model-based techniques enables the development of safe, informative, and subject-tailored clinical tests for model identification with limited experimental effort [27].

The current study has several limitations. First, the sample size was small, and there was a heterogeneous distribution among patient groups. We assessed only the immediate impacts of GVS on vertigo and imbalance by estimating the output variables right after each GVS application, but we did not examine any long-term effects of GVS. Our three output measures, D-VAS, ABC, and SARA scores, are conventional behavioral scales based on structured questionnaires. Their ability to precisely capture changes within 30 min post-stimulation is questionable. It is unclear whether a patient’s response to questions, such as their confidence in reaching an upper shelf, would genuinely change in such a short period. In addition, given the growing body of evidence demonstrating the effects of GVS on cognitive aspects [17,18,19,21], it should be considered that these effects on output estimations were associated with GVS-induced cognitive implications. Lastly, although we conducted fewer than three GVS sessions a day with at least 30 min intervals between them, the effect of GVS seems to be both profound and sustained if the GVS stimulus is repeated in multiple sessions [14]. Several variables can influence the duration of GVS effects, including the intensity and length of stimulation, as well as individual differences. Based on our observations and findings from other studies, the immediate effects of GVS—such as induced sway or sensations of dizziness—are generally short-lived, persisting for mere seconds to a few minutes. However, potential aftereffects or the residual impact of GVS can last longer, sometimes up to several hours. Given these variations, there is a pressing need for comprehensive, controlled studies focusing on the cumulative and long-term effects of GVS. Such research, ideally with a sizable sample, is crucial before integrating GVS into clinical practice and pinpointing the most efficacious paradigm.

To summarize, this study is the first to apply an optimal design method using clinical data to identify the most efficient GVS modes for patients with vestibular and cerebellar disorders. The method utilized three input parameters, including waveform, amplitude, and duration of stimulation, along with three output variables assessing vertigo and imbalance symptoms. This is of great clinical significance because the optimal GVS intervention regimen has yet to be established, despite the increasing use of GVS. Unlike the GVS threshold determination method, which only considers sensory perception, the design optimization method incorporates input and output clinical variables to determine the best GVS paradigm for direct interventional goals, such as improving vertigo and balance function. With growing evidence supporting the efficacy of GVS in enhancing balance and cognitive performance, our findings have important implications for maximizing the clinical application of GVS paradigms.

## Figures and Tables

**Figure 1 brainsci-13-01333-f001:**
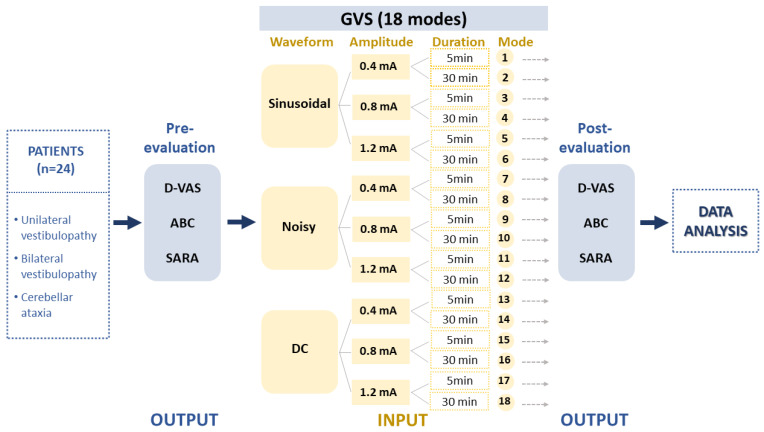
Experimental design. The study collected data from patients with unilateral (UVD) and bilateral vestibulopathy (BVD) and cerebellar ataxia (CA) to evaluate the effects of 18 different GVS modes. The GVS modes were determined by three independent input parameters, including waveforms, amplitudes, and application durations. The study then analyzed the pre–post data using the D-VAS, ABC, and SARA indices as output variables. GVS = galvanic vestibular stimulation; DC = direct current; UVD = unilateral vestibular deafferentation; BVP = bilateral vestibulopathy; CA = cerebellar ataxia; MSA-C = multiple system atrophy, cerebellar subtype; D-VAS = Dizziness Visual Analogue Scale; ABC = Activities-specific Balance Confidence Scale; SARA = Scale for Assessment and Rating of Ataxia.

**Figure 2 brainsci-13-01333-f002:**
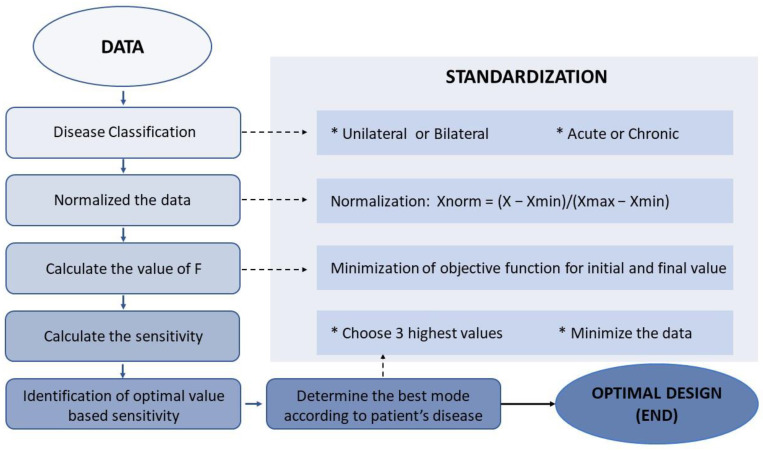
The flowchart outlines the steps for processing data and assessing model performance. Xmin, Xmax, and Xnorm are used to represent the minimum, maximum, and normalized values of each variable in all data sets.

**Table 1 brainsci-13-01333-t001:** Demographic characteristics and vestibular function tests in patients with vestibulopathy and cerebellar disorders.

Pt.	Age	Sex	Diagnosis	Time from the Onset	Classification	MMSE	Caloric Paresis (%, Side)	vHIT Gain		oVEMP AR, %	cVEMP AR, %
								Right HC	Left HC		
1	54	F	Vestibular neuritis, L	3 days	Acute UVP	29	29.54	1.08	0.52	12	1
2	51	M	Vestibular neuritis, L	3 days	Acute UVP	30	22.67	0.62	0.27	11	9
3	59	M	Vestibular neuritis, L	4 days	Acute UVP	30	84.39 (L)	0.94	0.26	52	47
4	64	F	Vestibular neuritis, L	3 days	Acute UVP	29	48.08 (L)	1.21	0.76	32	30
5	57	M	Vestibular neuritis, R	3 days	Acute UVP	29	55.49 (R)	0.34	0.9	43	12
6	31	M	Vestibular neuritis, R	3 days	Acute UVP	30	65.3 (R)	0.85	0.97	23	6
7	59	M	Vestibular neuritis, L	5 days	Acute UVP	29	39.44 (R)	1.07	1.13	18	21
8	82	M	Chronic UVP, R	34 months	Chronic UVP	28	34.55 (R)	0.41	0.36	34	31
9	72	F	Chronic UVP, R	71 months	Chronic UVP	28	36.07 (R)	0.56	1.14	21	37
10	49	M	Bilateral vestibulopathy	52 months	Chronic BVP	30	12.3	0.36	0.31	19	41
11	26	M	Bilateral vestibulopathy	61 months	Chronic BVP	30	13.03	0.16	0.16	7	2
12	71	M	Presbyvestibulopathy	42 months	Chronic BVP	29	10.01	0.44	0.59	n/a	n/a
13	68	M	Presbyvestibulopathy	48 months	Chronic BVP	28	21	0.30	0.47	n/a	n/a
14	63	M	Presbyvestibulopathy	69 months	Chronic BVP	30	15.2	0.45	0.39	21	31
15	71	M	CANVAS	65 months	Chronic BVP	29	18.42	0.21	0.19	15	30
16	59	F	CANVAS	44 months	Chronic BVP	30	20.1	0.28	0.34	n/a	n/a
17	72	F	CANVAS	57 months	Chronic BVP	27	17.91	0.18	0.24	26	31
18	61	M	CANVAS	46 months	Chronic BVP	30	20.0	0.37	0.40	11	7
19	59	M	MSA-C	80 months	CA	29	14.58	1	0.91	11	11
20	57	M	MSA-C	90 months	CA	30	15.5	0.91	0.90	8	4
21	69	M	MSA-C	67 months	CA	27	4.61	0.95	0.93	4	23
22	22	M	SCA type 2	23 months	CA	30	19.77	1.02	1.04	28	24
23	23	M	SCA type 2	25 months	CA	30	19.77	1.02	1.04	38	24
24	56	M	SCA type 6	15 months	CA	27	2.46	1.09	1.16	0	13
25	42	M	Cerebellar ataxia *	63 months	CA	30	19.7	1.07	1.02	6	4
26	55	M	Cerebellar ataxia ^†^	67 months	CA	28	19.06 (L)	0.98	1.03	25	8
27	58	M	Cerebellar ataxia ^‡^	62 months	CA	28	8.08	0.91	0.94	36	15
28	73	M	Late-onset cerebellar ataxia	78 months	CA	27	13.5	0.89	0.91	3	8
29	62	M	Late-onset cerebellar ataxia	97 months	CA	30	9.5	1.02	0.99	11	8
30	61	M	Late-onset cerebellar ataxia	10 months	CA	30	−27.11	0.74	0.81	6	4
31	62	M	Late-onset cerebellar ataxia	13 months	CA	29	21.6	1.12	1.09	56	8

Pt. = patient; MMSE = Mini–Mental State Examination; F = female; M = male; L = left; R = right; UW = unilateral weakness; vHIT = video Head Impulse Test; hVOR = horizontal vestibulo-ocular reflex; oVEMP AR = n10 amplitude asymmetry ratio; cVEMP AR = p13 amplitude ratio; UVP = Unilateral vestibulopathy; BVP = bilateral vestibulopathy; PD = Parkinson’s disease; CA = cerebellar ataxia; CANVAS = cerebellar ataxia neuropathy and vestibular areflexia syndrome; MSA-C = multiple system atrophy, cerebellar subtype; SCA = spinocerebellar ataxia; n/a = not available. * heterozygous mutations in the KCNC3 gene (c.1746_1754del and p.Pro583_Pro585del & c.1738C>A and p.Pro580Thr), SLC52A2 gene (c.204G>A and p.Trp68Cys), WFS1 (c.1526T>G and p.Val509Gly) gene, and PNKP gene (c.949C>A and p.Leu317Ile) were identified. ^†^ heterozygous mutations in the SACS gene (c.1066A>G and p.Ile356Val & c.7223A>G and p.Asp2408Gly) and SYNE1 gene (c.2617A>G and p.Lys873Glu) were identified. ^‡^ heterozygous mutations in the CACNA2D2 gene (c.502G>A and p.Ala168Thr) and COQ8A gene (c.241T>C and p.Phe81Leu) were identified.

**Table 2 brainsci-13-01333-t002:** Ranking of the estimated total F value of GVS efficacy in patients with vestibulopathies.

Rank	GVS Protocol				Min F	Sum of the F Value	Sensitivity			Vestibular Perception (Spinning, Tilting or Tingling Sensation)
	Mode Number	Waveform	Amplitude (mA)	Duration (min)			D-VAS	ABC	SARA	
Unilateral vestibulopathy (n = 9)										
1	7	Noisy	0.4	30	−0.65	−0.8	2.13	−9.51	2.35	no
2	1	Sinusoidal	0.4	30	−0.39	−0.69	1.58	−7.38	4.32	no
3	16	DC	0.8	5	−0.41	−0.64	2.56	−4.26	2.67	yes (spinning)
4	2	Sinusoidal	0.4	5	−0.29	−0.54	2.16	−10.8	2.25	no
5	10	Noisy	0.8	5	−0.29	−0.49	1.57	−3.55	12.27	no
6	14	DC	0.4	5	−0.21	−0.45	3.57	−1.85	5.57	no
7	9	Noisy	0.8	30	−0.34	−0.41	6.63	−5.76	1.48	no
8	13	DC	0.4	30	−0.13	−0.39	1.26	−9.69	9.83	no
9	4	Sinusoidal	0.8	5	−0.17	−0.31	1.9	−2.91	7.71	no
10	8	Noisy	0.4	5	−0.12	−0.3	1.58	−2.72	n.c.	no
11	12	Noisy	1.2	5	−0.28	−0.28	n.c.	n.c.	1	no
12	5	Sinusoidal	1.2	30	−0.19	−0.25	1.33	−4	n.c.	no
13	3	Sinusoidal	0.8	30	−0.07	−0.05	n.c.	−4.26	1.31	no
14	17	DC	1.2	30	−0.09	−0.09	0.68	n.c.	−2.12	yes (tilting, tingling)
15	15	DC	0.8	30	−0.16	−0.08	−0.45	−0.76	0.53	no
16	18	DC	1.2	5	−0.04	−0.04	n.c.	n.c.	1	yes (tingling)
17	6	Sinusoidal	1.2	5	−0.01	0.05	n.c.	−1	n.c.	no
18	11	Noisy	1.2	30	−0.08	0.08	0.45	3.67	−1.06	no
Bilateral vestibulopathy (n = 9)										
1	9	Noisy	0.8	30	−0.38	−0.38	1.75	−2.33	n.c.	no
2	7	Noisy	0.4	30	−0.14	−0.22	n.c.	−1	n.c.	no
3	2	Sinusoidal	0.4	5	−0.01	−0.01	1	n.c.	n.c.	no
4	6	Sinusoidal	1.2	5	−0.13	−0.13	1	n.c.	n.c.	no
5	8	Noisy	0.4	5	0	0	n.c.	n.c.	n.c.	no
6	11	Noisy	1.2	30	0	0	n.c.	n.c.	n.c.	no
7	4	Sinusoidal	0.8	5	0	0	n.c.	n.c.	n.c.	no
8	5	Sinusoidal	1.2	30	0	0	n.c.	n.c.	n.c.	no
9	1	Sinusoidal	0.4	30	0	0	n.c.	n.c.	n.c.	no
10	12	Noisy	1.2	5	0	0.02	n.c.	−1	n.c.	no
11	10	Noisy	0.8	5	0	0.09	n.c.	−1	n.c.	no
12	3	Sinusoidal	0.8	30	0	0.4	1.6	−2.67	n.c.	no

UVP = unilateral vestibulopathy (n = 9); BVP = bilateral vestibulopathy (n = 9); GVS = galvanic vestibular stimulation; DC = direct current; D-VAS = Dizziness Visual Analogue Scale; ABC = Activities-specific Balance Confidence Scale; SARA = Scale for Assessment and Rating of Ataxia; n.c. = there was no change in scale between “before and after” GVS intervention.

**Table 3 brainsci-13-01333-t003:** Ranking of the estimated total F value of GVS efficacy in cerebellar ataxia (n = 13).

Rank	GVS Protocol				Min F	Sum of the F Value	Sensitivity			Vestibular Perception (Spinning, Tilting or Tingling Sensation)
	Mode	Waveform	Amplitude (mA)	Duration (min)			D-VAS	ABC	SARA	
1	10	Noisy	0.8	5	−0.94	−0.98	1.3	−6.66	12.21	no
2	7	Noisy	0.4	30	−0.38	−0.88	1	16.6	15.69	no
3	12	Noisy	1.2	5	−0.54	−0.63	1.01	−126	n.c.	no
4	8	Noisy	0.4	5	−0.22	−0.42	1.67	−3.26	10.46	no
5	9	Noisy	0.8	30	−0.38	−0.32	1.01	−97.15	n.c.	no
6	2	Sinusoidal	0.4	5	−0.08	−0.21	n.c.	−1.24	5.22	no
7	4	Sinusoidal	0.8	5	−0.2	−0.2	1.64	n.c.	2.56	no
8	6	Sinusoidal	1.2	5	−0.13	−0.18	1.45	−3.24	n.c.	no
9	11	Noisy	1.2	30	−0.13	−0.01	n.c.	−1	n.c.	no
10	5	Sinusoidal	1.2	30	−0.03	−0.01	n.c.	−1	n.c.	no
11	3	Sinusoidal	0.8	30	−0.04	0.07	n.c.	−2.29	1.78	no
12	1	Sinusoidal	0.4	30	0	0.1	1.56	−2.79	n.c.	no

GVS = galvanic vestibular stimulation; DC = direct current; D-VAS = Dizziness Visual Analogue Scale; ABC = Activities-specific Balance Confidence Scale; SARA = Scale for Assessment and Rating of Ataxia; n.c. = there was no change in scale between “before and after” GVS intervention.

## Data Availability

All individual data on the participants that underlie the results reported in this article, after de-identification (manuscript, tables, and figures), will be shared.

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
