# Peer review of "Optimal Design of Galvanic Vestibular Stimulation for Patients with Vestibulopathy and Cerebellar Disorders"

_brainsci, 2023, doi:10.3390/brainsci13091333_

Round 1
Reviewer 1 Report
The work is well executed. It represents an attempt to optimize the characteristics of the galvanic stimulus to obtain the best results. This was achieved through the application of the optimization design used in identifying the effectiveness of drugs. With this method it was possible to identify which form of electrical stimulus, amplitude and duration was most effective in reducing postural and perceptual alterations of subjects with vestibular and cerebellar alterations. The number of observations is not large, but the work shows how it is possible to optimize galvanic stimuli by using this method. All this is limited to aspects related to the vestibular system and balance. The same method should also be applied to higher functions, in which the galvanic stimulus looks induce better performance. I have not found critical points to report, although I would like an explanation of the reasons why continuous or sinusoidal or noisy stimuli are differently effective in neuronal activation in relation to their intensity and duration.
Author Response
Thank you for your comments. In a nutshell, in our study, for patients with vestibulopathy and cerebellar disorders, GVS modes using subthreshold currents (0.4-0.8 mA) for extended periods (30 minutes) tend to be more effective than those employing suprathreshold currents (1.2 mA) over shorter intervals (5 minutes). However, a direct comparison remains elusive, primarily since our study is the first to apply the 'optimal design' methodology. This technique encompasses a plethora of parameters to assess and prioritize GVS efficacy, and thus, pinpointing congruent results for a direct comparison has proven challenging.
To pinpoint the optimal combinations, referred to as the "best triads", derived from the matrix of waveform, intensity (amplitude), and duration, the biophysiological effects of each triad should be thoroughly investigated including the functional neuroimaging techniques. To date, the interplay between the effectiveness of GVS and each waveform, intensity, and duration, remains uncharted territory. Our hypothesis posits that each triad uniquely amplifies a synergistic influence on the modulation of vestibular afferent firing rates. In terms of intensity, stronger DC currents will cause a more pronounced initial shift in neural activity. However, continuous stimulation (DC) can lead to a steady-state activation or adaptation of the vestibular nerve fibers. After an initial change in activity, the neurons may adapt, reducing their firing rate over time even if the stimulus continues. While the intensity and frequency of the sinusoid will directly influence which neuronal populations are activated. Notably, irregular neurons tend to have a lower activation threshold than their regular counterparts, as well as different resonance frequencies. The duration of the stimulus defines how long this neuronal modulation lasts, and it's believed that adaptation may be slower compared to DC stimulation. Given what we currently know, we speculate that each triad has its own unique combined effect on the modulation of both irregular and regular vestibular afferent firing rates. Such a synergistic impact is deemed beneficial if it's potent enough to cause meaningful charge modulation in the vestibular cortices, thus inducing a neurobiological effect. At the same time, it's crucial to avoid excessive charge buildup in the tissues, as this could be detrimental and potentially cause neuronal damage.
As the reviewer’s comment, we added this issue in the revised manuscript (pages 15-16).

Reviewer 2 Report
Thank you for the opportunity to provide peer review of your article titled “Optimal Design of Galvanic Vestibular Stimulation for Patients with Vestibulopathy and Cerebellar Disorders”. I believe this is an interesting study with clinical implication. However, I have pointed out the below areas that need to be addressed. Furthermore, I recommend that this manuscript undergo sufficient statistical review, as I am not comfortable providing sufficient feedback on the approaches used. I have highlighted the areas requiring this review below as well.
Abstract:
The abstract is well written and provides all necessary information.
Introduction:
The introduction does a good job of introducing GVS and discussing the implications of its use.
What is needed is a description of why this is also useful in Cerebellar Disorders. The vestibular elements of the stimulation are discussed, but including information about Cerebellar disorders and the usefulness of GVS should be introduced to the reader.
Methods:
Why was this study designed to include cerebellar disorders along with unilateral and bilateral vestibular disorders? By including a much more heterogeneous sample like those individuals living with cerebellar dysfunction seems like it greatly complicates a much cleaner analysis that could be provided by restricting the study to only unilateral and bilateral vestibular dysfunction.
A major concern is the amount of time between the end of GVS stimulation and completion of the outcome measures? Was this controlled for? If so it needs to be indicated. If it was not controlled this is a major flaw in the study, duration of time between the stimulation and completion of the outcome measures seems to be a major factor in performance.
What position were participants in during delivery of the GVS. Was this always the same? Why was this position chosen. Provide this information please.
One major issue that needs to be addressed in the methods is the rationale for which diagnoses received which of the GVS parameters. It is described which diagnoses received which parameters, but no rationale as to why these decisions were made is provided. It needs to be made clear to the reader as to why these decisions were made. Furthermore, the rationale for providing the various stimulations must be supported by the literature or previous data. As it is currently presented it is confusing.
Need for statistical review is needed in multiple places in the methods and this has been recommended to the editor:
1) Methods used for data output normalization and transformation should be reviewed by statistical reviewer
2) Data integration, ranking and ultimately the sensitivity approach should be reviewed by a statistical reviewer.
Results:
The authors mention that “a few” people reported sensations of spinning or tingling over the mastoid process. This is not sufficient to say “a few” data should be provided regarding the number of people from each group who experienced these sensations, for how long, and if they altered the ability to continue with the GVS delivery. This is a key component in understanding the clinical utility of GVS. This can be elaborated upon in the text and may also be presented in a table.
Did the participants provide any rating of dizziness, balance, or other symptoms during stimulation? If so please provide this data.
Similar to what was mentioned in review of the methods. The average time to completing the outcome measures should be provided, as this length of time seems to be a critical component of response.
Were any functional measures collected? For example do you have balance data that could be presented in response to the GVS stimulation? This would greatly strengthen the conclusion that these parameters were optimum for these conditions.
Discussion:
When describing the findings in the first paragraph language such as “the best option” is not ideal. Rather the authors should use language describing how each of the parameters performed… e.g., DC GVS with XX parameters was found to perform best in our analysis when used in people with UVP. This type of language should be used throughout. When using worlds like best option that indicates various things to the readers, such as performance of the intervention, but also patient preference, etc.
The authors need to describe how the optimized parameters from this study compare to outcomes on the same or similar measures from other GVS studies. Are the improvements found here the same or different, or better than what has been previously published?
Author Response
Thank you for the opportunity to provide peer review of your article titled “Optimal Design of Galvanic Vestibular Stimulation for Patients with Vestibulopathy and Cerebellar Disorders”. I believe this is an interesting study with clinical implication. However, I have pointed out the below areas that need to be addressed. Furthermore, I recommend that this manuscript undergo sufficient statistical review, as I am not comfortable providing sufficient feedback on the approaches used. I have highlighted the areas requiring this review below as well.
Abstract:
The abstract is well written and provides all necessary information.
Introduction:
The introduction does a good job of introducing GVS and discussing the implications of its use.
What is needed is a description of why this is also useful in Cerebellar Disorders. The vestibular elements of the stimulation are discussed, but including information about Cerebellar disorders and the usefulness of GVS should be introduced to the reader.
Reply:
Thank you for your valuable input. We have integrated the following statement on page 4: “Given the substantial afferent and efferent connections that exist between the vestibular nuclei and the cerebellum, particularly the posterior cerebellum, along with emerging evidence of the positive effects of non-invasive cerebellar stimulation in treating cerebellar disorders, there is a theoretical basis for suggesting that GVS could potentially enhance cerebellar function and address cerebellar dysfunctions.”
Methods:
Why was this study designed to include cerebellar disorders along with unilateral and bilateral vestibular disorders? By including a much more heterogeneous sample like those individuals living with cerebellar dysfunction seems like it greatly complicates a much cleaner analysis that could be provided by restricting the study to only unilateral and bilateral vestibular dysfunction.
Reply:
Thank you for raising those points. We agree with the reviewer’s concern. And we depicted several limitations of the current study including the small sample size and a heterogeneous distribution among patient groups (pages 17-18).
Optimal design, however, maximizes information yield while taking into account budget constraints that limit the resources available for the study, and has recently emerged as a valuable statistical methodology for designing a wide range of studies. It provides a systematic, quantitative approach to selecting study units (i.e., patient groups) in the most informative manner for observational studies and assigning study units to intervention conditions in the most informative manner for experimental studies. This novel approach has been widely used in clinical research, particularly in studies exploring a typical pattern of pharmacokinetics over time, which has assisted in reducing the number of sampling times, improving existing therapies or diagnostics, and providing recommendations for effective dose regimen (pages 5-6).
The optimal design methodology enables us to deduce specific GVS parameters tailored to individual disorders with a relatively small sample size. Our intention in the current study, is to apply this approach to not only patients with vestibulopathy but also to those suffering from cerebellar disorders. As the reviewer’s comment, we are now planning to recruit more patients in the each group.
A major concern is the amount of time between the end of GVS stimulation and completion of the outcome measures? Was this controlled for? If so it needs to be indicated. If it was not controlled this is a major flaw in the study, duration of time between the stimulation and completion of the outcome measures seems to be a major factor in performance.
Reply:
Thank you for your recommendation. Each assessment session commenced five minutes after the cessation of the stimulation and typically lasted between 10 to 15 minutes. We added this in the page 7 of the revised manuscript.
What position were participants in during delivery of the GVS. Was this always the same? Why was this position chosen. Provide this information please.
Reply:
Thank you for your recommendation. On page 7 of the revised manuscript, we have included the following: 'Participants were seated in a comfortable chair equipped with armrests, located in a soundproof and dimly lit room.
One major issue that needs to be addressed in the methods is the rationale for which diagnoses received which of the GVS parameters. It is described which diagnoses received which parameters, but no rationale as to why these decisions were made is provided. It needs to be made clear to the reader as to why these decisions were made. Furthermore, the rationale for providing the various stimulations must be supported by the literature or previous data. As it is currently presented it is confusing.
Reply:
Thank you for your recommendation. In this study, from the multitude of possible combinations of waveform, amplitude, and duration, we opted for 18 modes as a pragmatic approach for the initial exploration in a clinical context (Figure 1). We used three waveforms: DC, sinusoidal, and noisy. Regarding amplitude, we took into account the vestibular perceptual threshold of GVS. Although there isn't a standardized threshold, many clinical studies accept a GVS threshold of 1mA. Therefore, we chose amplitudes of 0.4 mA (subthreshold), 0.8 mA (around threshold), and 1.2 mA (suprathreshold). We also utilized short and long GVS durations: 5 minutes and 30 minutes, respectively. However, patients with UVD were exposed to all 18 modes, while those with BVD or cerebellar ataxia were given only 12 modes, excluding the DC current. The DC mode may not be apt for bilateral conditions like BVD or cerebellar ataxia due to its polarization effects. The mode order was randomized using Microsoft Excel's =Rand() function. We have incorporated this information on page 8 of the revised manuscript.
Need for statistical review is needed in multiple places in the methods and this has been recommended to the editor:
1) Methods used for data output normalization and transformation should be reviewed by statistical reviewer
2) Data integration, ranking and ultimately the sensitivity approach should be reviewed by a statistical reviewer.
Reply:
Per the reviewer's recommendation, we've expanded on the research methods in the revised manuscript (pages 9-11).
Results:
The authors mention that “a few” people reported sensations of spinning or tingling over the mastoid process. This is not sufficient to say “a few” data should be provided regarding the number of people from each group who experienced these sensations, for how long, and if they altered the ability to continue with the GVS delivery. This is a key component in understanding the clinical utility of GVS. This can be elaborated upon in the text and may also be presented in a table. Did the participants provide any rating of dizziness, balance, or other symptoms during stimulation? If so please provide this data.
Reply:
Thank you for your suggestion. A handful of patients described sensations of spinning or tilting, as well as mild tingling at the mastoid processes. These sensations were observed primarily at higher DC stimulation levels (1.2 mA) but subsided after discontinuing the stimulation. These findings are incorporated in Tables 2, 3, and also in Supplementary Table 1.
Similar to what was mentioned in review of the methods. The average time to completing the outcome measures should be provided, as this length of time seems to be a critical component of response.
Reply:
Each assessment session commenced five minutes after the cessation of the stimulation and typically lasted between 10 to 15 minutes.
Were any functional measures collected? For example do you have balance data that could be presented in response to the GVS stimulation? This would greatly strengthen the conclusion that these parameters were optimum for these conditions.
Reply:
In response to the reviewer's comments, we tried to ascertain the best GVS parameters for patients with vestibulopathy and cerebellar disorders. Leveraging optimal design calculations, we delved into the effects of GVS on postural balance and its vestibular influence. In our study, we selected clinical assessments, which capture perceptions of dizziness and imbalance. These assessments include the Dizziness-Visual Analogue Scale (D-VAS), Activities-specific Balance Confidence scale (ABC), and the Scale for Assessment and Rating for Ataxia (SARA), which served as our output variables for the optimal design calculation (Figure 1). Additionally, we incorporated a sensitivity analysis (SA) to identify pivotal design parameters.
Earlier research has extensively described the use of functional measurements to evaluate the influence of GVS on balance [Iwasaki et al. (2014), Iwasaki et al. (2018), Fujimoto et al. (2016), Fujimoto et al. (2018), Yamamoto et al. (2005), Wuehr et al. (2016)]. We've addressed this topic in the revised manuscript.
Discussion:
When describing the findings in the first paragraph language such as “the best option” is not ideal. Rather the authors should use language describing how each of the parameters performed… e.g., DC GVS with XX parameters was found to perform best in our analysis when used in people with UVP. This type of language should be used throughout. When using worlds like best option that indicates various things to the readers, such as performance of the intervention, but also patient preference, etc.
Reply:
Thank you for your invaluable suggestions. Based on the detailed recommendations from the reviewer, we have revised the paragraphs in the Discussion section.
From our analysis, patients with UVP exhibited the best outcomes when subjected to either noisy or sinusoidal GVS at an amplitude of 0.4 mA for 30 minutes. Another promising approach for UVP patients involves the use of DC GVS, with the cathode aligned to the lesion side and an amplitude of 0.8 mA for a brief period of 5 minutes (Table 2). In the case of BVP patients, noisy GVS, particularly with amplitudes of 0.8 or 0.4 mA over 30 minutes, was identified as the most effective. Subsequently, sinusoidal GVS at a 0.4 mA amplitude for 5 minutes also demonstrated encouraging results. For patients diagnosed with cerebellar ataxia, our findings prominently feature noisy GVS. When applied at amplitudes of 0.8 or 0.4 mA for either 5 or 30 minutes, it produced the most favorable outcomes ( Table 3).
The authors need to describe how the optimized parameters from this study compare to outcomes on the same or similar measures from other GVS studies. Are the improvements found here the same or different, or better than what has been previously published?.
Reply:
Thank you for your insightful recommendation. We have made efforts to correlate our observations with findings from earlier studies in the section discussing how 'The effects of GVS are stimulus-dependent.' Notably, the challenge remains in determining how to fine-tune the stimulation parameters to amplify beneficial outcomes. From our data, GVS modes employing subthreshold currents ranging from 0.4 to 0.8 mA for extended durations (30 minutes) seem more promising than those using suprathreshold currents (1.2 mA) for shorter spans (5 minutes), especially for patients with vestibulopathy and cerebellar disorders.
However, the 'optimal design' methodology encompasses a plethora of parameters to assess and prioritize GVS efficacy, and thus, pinpointing congruent results for a direct comparison has proven challenging. To pinpoint the optimal combinations, the biophysiological effects of each triad should be thoroughly investigated including the functional neuroimaging techniques. Therefore, such research, ideally with a sizable sample, is crucial before integrating GVS into clinical practice and pinpointing the most efficacious paradigm.

Reviewer 3 Report
The paper deals with important topic and uses up to date statistical methodology for its experimental design.
The results have potential impact on clinical practice. However there are several flaws, that should be dealt.
First of all, the study group is small a heterogenous. It should be clearly explained in the discussion by which mechanisms so pathpphysiologically different conditions like is labyrinthine failure and cerebellar degeneration cpould be treated by the same procedure.
The ABC score, used as one of three output measures is a typical behavioural scale and it it not clear, why the patient should respond differently after 30 minutes of stimulation on the question, e.g. how confident he/she is on reaching on upper shelf, if there was no opportunity to test this situation. This type of scale is not well chosen for the situation in which immediate and short lasting should be measured.
Another problem is that subject were stimulated with 2-3 different setup during one day. It is not clear, how big the carry over effect could be form the preceding session to the next. This can influence the effect quite strongly.
These objections should be dealt in detail in the discussion.
Also it has to be stated, that this is a methodological study, which has shown short lasting beneficial symptomatic effect in patient with vertigo of various origin. The results needs to to be reproduced in large groups controlled by placebo.
Author Response
The paper deals with important topic and uses up to date statistical methodology for its experimental design. The results have potential impact on clinical practice. However there are several flaws, that should be dealt.
First of all, the study group is small a heterogenous. It should be clearly explained in the discussion by which mechanisms so pathophysiologically different conditions like is labyrinthine failure and cerebellar degeneration could be treated by the same procedure.
Reply:
Thank you for your insightful recommendation. We agree with the reviewer’s concern. And we depicted several limitations of the current study including the small sample size and a heterogeneous distribution among patient groups (pages 17-18).
Optimal design, however, maximizes information yield while taking into account budget constraints that limit the resources available for the study (reducing the sample size), and has recently emerged as a valuable statistical methodology for designing a wide range of studies. The optimal design methodology enables us to deduce specific GVS parameters tailored to individual disorders with a relatively small sample size.
In our study, we aim to optimize GVS parameters for both vestibulopathy and cerebellar disorder patients, despite their differing pathophysiology but similar symptoms of imbalance and dizziness. While numerous published reports have demonstrated the beneficial effects of GVS on balance performance in vestibulopathy also in cerebellar disorders, fundamental questions remain regarding the optimal GVS parameters.
This issue has been incorporated in the revised manuscript (page 5) following the reviewer’s suggestion.
The ABC score, used as one of three output measures is a typical behavioural scale and it it not clear, why the patient should respond differently after 30 minutes of stimulation on the question, e.g. how confident he/she is on reaching on upper shelf, if there was no opportunity to test this situation. This type of scale is not well chosen for the situation in which immediate and short lasting should be measured.
Reply:
Thank you for your invaluable suggestions. Based on the detailed recommendations from the reviewer, we have revised the paragraphs in the limitation of the study section (page 17).
Our three output measures, the D-VAS and ABC scores, are conventional behavioral scales based on structured questionnaires. Their ability to precisely capture changes within 30 minutes post-stimulation is questionable. It's unclear whether a patient's response to questions, such as their confidence in reaching an upper shelf, would genuinely change in such a short period.
Another problem is that subject were stimulated with 2-3 different setup during one day. It is not clear, how big the carry over effect could be form the preceding session to the next. This can influence the effect quite strongly.
Reply:
Thank you for your concerns. The effect of GVS seems to be both profound and sustained if the GVS stimulus is repeated in multiple sessions. Several variables can influence the duration of GVS effects, including the intensity and length of stimulation, as well as individual differences. Based on our observations and findings from other studies, the immediate effects of GVS—such as induced sway or sensations of dizziness—are generally short-lived, persisting for mere seconds to a few minutes. However, potential aftereffects or the residual impact of GVS can last longer, sometimes up to several hours. Given these variations, there's a pressing need for comprehensive, controlled studies focusing on the cumulative and long-term effects of GVS. Such research, ideally with a sizable sample, is crucial before integrating GVS into clinical practice and pinpointing the most efficacious paradigm.
We have incorporated this limitation on page 18 of the revised manuscript.
These objections should be dealt in detail in the discussion.
Also it has to be stated, that this is a methodological study, which has shown short lasting beneficial symptomatic effect in patient with vertigo of various origin. The results needs to be reproduced in large groups controlled by placebo..
Reply:
We appreciate your valuable recommendation.

Round 2
Reviewer 3 Report
Thank you for your replies and changes made in the paper!